# Molecular Characterization of Alkyl Nitrates in Atmospheric Aerosols by Ion Mobility Mass Spectrometry

Xuan Zhang[1,*], Haofei Zhang[2,3], Wen Xu[4], Xiaokang Wu[5], Geoffrey S. Tyndall[1], John J. Orlando[1], John T. Jayne[4], Douglas R. Worsnop[4], and Manjula R. Canagaratna[4,*]

[1] Atmospheric Chemistry Observation & Modeling Laboratory, National Center for Atmospheric Research, Boulder, CO 80301, USA

[2] Department of Chemistry, University of California, Riverside, CA 92521, USA

[3] Environmental Toxicology Program, University of California, Riverside, CA 92521, USA

[4] Center for Aerosol and Cloud Chemistry, Aerodyne Research Inc., Billerica, MA 01821, USA

[5] Department of Atmospheric Sciences, Texas A&M University, College Station, TX 77843, USA

*Correspondence to*: Xuan Zhang (xuanz@ucar.edu)
Manjula R. Canagaratna (mrcana@aerodyne.com)

**Abstract**
We demonstrate the capability of the Ion Mobility Mass Spectrometry (IMS-MS) for
molecular characterization of reactive and short-lived alkyl nitrates (ANs) in atmospheric
aerosols. We show significantly enhanced sensitivity towards the intact molecules of ANs
by ultimately two orders of magnitude with the addition of inorganic anions such as
chloride and nitrate to the negative electrospray to promote the ion adduct formation.
This approach enables the measurement of ANs that have low tendency to form
molecular ions on their own with improved limit of detection in the range of 0.1 to 4.3
μM. Molecular identities of the ANs are well constrained by the developed collision
cross section vs. mass to charge ratio correlation, which provides a two-dimensional
separation of the $-ONO_2$ containing compounds on the basis of their molecular size and
geometry. Structural information of the nitrate molecules is further probed by the
identification of characteristic fragments produced from the collision induced
dissociation of parent AN adducts. Application of the IMS-MS technique is exemplified
by the identification of hydroxy nitrates in secondary organic aerosols produced from the
photochemical oxidation of isoprene.





## 1. Introduction

Alkyl nitrates (ANs; ANs = $RONO_2$) constitute a major fraction and serve as a temporary reservoir of total reactive nitrogen oxides in the atmosphere (Perring et al., 2013). ANs are primarily produced from the hydroxyl radical (OH) initiated oxidation of volatile organic compounds (VOCs) in the presence of nitrogen oxides ($NO_x$) during daytime and the nitrate radical ($NO_3$) initiated oxidation of alkenes during nighttime. Once formed, ANs are primarily subjected to further chemical transformation leading to the recycling of $NO_x$, partitioning into the particle phase forming secondary organic aerosols (SOA), or deposition resulting in the loss of atmospheric $NO_x$. Characterization of alkyl nitrates is of crucial importance in understanding regional $NO_x$ budget, tropospheric ozone production, as well as chemical mechanisms leading to the SOA formation (Brown et al., 2009; Farmer et al., 2011; Rollins et al., 2012; Rosen et al., 2004).

A suite of analytical techniques, such as thermal dissociation laser-induced-fluorescence spectroscopy (TD-LIF) (Thornton et al., 2000; Day et al., 2002; Wooldridge et al., 2010), chemical ionization mass spectrometry (CIMS) (Beaver et al., 2012; Loza et al., 2014; Krechmer et al., 2015; Nguyen et al., 2015; Schwantes et al., 2015; Teng et al., 2015; Xiong et al., 2015; Schwantes et al., 2017b; Lambe et al., 2017), and gas chromatography coupled with electron capture detection (GC-ECD) (Atlas, 1988; O'Brien et al., 1995; He et al., 2011), have been employed for *in situ* measurement of total and individual ANs in the gas phase. Observations of ANs in the particle phase, however, are rather limited due to the intensive denitrification during the preparation and analysis of particle samples. Efforts have been made to characterize the total amount of ANs and the number of $-ONO_2$ functional groups using TD-LIF and Fourier transform infrared spectroscopy (FTIR) (Rollins et al., 2010; Russell et al., 2011). The $NO_2^+ / NO^+$ ratio derived from the aerosol mass spectrometry (AMS) measurements has also been used as an indicator for the presence of alkyl nitrates in submicrometer particles (Farmer et al., 2010; Kiendler-Scharr et al., 2016; Xu et al., 2017; Xu et al., 2018). These techniques have provided important insights into the prevalence and abundance of ANs in atmospheric aerosols, although the molecular information of individual ANs is lacking. Recent development on the filter inlet for gases and aerosols (FIGAERO) interfaced with the CIMS instrument has allowed for on-line speciation and quantification of functionalized alkyl nitrates in the particle phase (Lee et al., 2016). While the molecular composition of any given compounds can be inferred from the mass spectra, structural

information on isomeric and isobaric species that are commonly produced from
atmospheric chemical transformation is not available from CIMS measurements.
In this study, we present the first demonstration of the Ion Mobility Mass
Spectrometry (IMS-MS) interfaced with an Electrospray Ionization (ESI) source that
enables the molecular characterization of alkyl nitrates in atmospheric aerosols. The IMS
technique has been widely employed in the fields of biochemistry and homeland security.
The majority of previous studies that adapted ESI for IMS analysis employed either the
Desorption Electrospray Ionization (DESI) to detect trace amounts of ANs on ambient
surfaces (Cotte-Rodríguez et al., 2005; Popov et al., 2005; Takáts et al., 2005; Justes et
al., 2007) or the Secondary Electrospray Ionization (SESI) for gas-phase ANs
measurements (Tam and Hill, 2004; Martínez-Lozano et al., 2009; Crawford and Hill,
2013). The analysis of ANs directly from liquid solutions, on the other hand, has not yet
been widely explored. Hilton et al. (2010) found that the $NO_3^-$ fragment dominates the
IMS spectra of several types of ANs measured in the negative ESI, suggesting these
nitrate molecules readily fragment due to the thermally labile nature of the -ONO$_2$
functionality, thereby resulting in the loss of molecular information of the targeted
compounds. Here we show that with the addition of selected anions including chloride,
nitrate, iodide, and acetate into the sprayed solution, molecular structures of ANs are
largely maintained by producing ion clusters of the form $[M+Cl]^-$, $[M+NO_3]^-$, $[M+I]^-$,
and $[M+Ac]^-$, respectively. The anion attachment represents a new option for the
detection of the $-ONO_2$ functionality that is unlikely to produce measurable amount of
molecular ions on its own during ESI. The optimal anion concentration to essentially
promote the ion adduct formation is on the order of milli-molar, which is significantly
higher than the level of those naturally present in ambient aerosols. We develop an
intrinsic collision cross section vs. mass to charge ratio correlation based on the ion
mobility measurements of five AN standards, providing a two-dimensional identification
of unknown molecules that are likely containing the $-ONO_2$ moiety. Additionally, the
molecular identity of ANs can be verified via the characteristic fragment produced from
the collision induced dissociation of the parent ion adducts. We apply the IMS-MS
technique to identify ANs in SOA produced from isoprene photochemistry.
**2. Experiments**
**2.1. Materials**
Organic nitrate and nitro standards stored in acetonitrile ampules, including 1-
mononitroglycerin (100 μg/mL, SigmaAldrich), 1,3-dinitroglycerin (100 μg/mL,
SigmaAldrich), pentaerythritol tetranitrate (1000 μg/mL, SigmaAldrich), hexahydro-
1,3,5-trinitro-1,3,5-triazine (1000 μg/mL, SigmaAldrich), and 2,4-dinitrotoluene (1000
μg/mL, SigmaAldrich), were further diluted with methanol (HPLC grade, J. T. Baker) to
5 μM or less for characterizing the performance of the Ion Mobility Mass Spectrometer.
Stock solutions of ammonium acetate (>99%, SigmaAldrich), ammonium chloride
(>99%, SigmaAldrich), sodium nitrate (>99%, SigmaAldrich), and sodium iodide (>99%,
SigmaAldrich) were prepared at a concentration of 10 mM in methanol. They were used
as additives at typical concentrations of $0.01 - 0.1$ mM in the ANs methanol solutions to
promote the ion adducts formation.

## 2.2. Experiments

SOA samples containing alkyl nitrates were generated from the OH-oxidation of
isoprene under high-$NO_x$ conditions in the NCAR 10 $m^3$ Atmospheric Simulation
Chamber (Zhang et al., 2018). $H_2O_2$ was used as the OH source by evaporating 133 μL
aqueous solution (30 wt% in water, SigmaAldrich) into the chamber with 5 L/min
purified air for ~120 min, resulting in a starting concentration of ~4 ppm (Wang et al.,
2009; He et al., 2010; Zhao et al., 2011; Cappa et al., 2013; Zhang and Seinfeld, 2013;
Schwantes et al., 2017a). Isoprene was injected into the chamber by evaporating ~17 μL
liquid standard (≥99%, SigmaAldrich) with 5 L/min purified air for ~20 min, resulting an
initial concentration of ~500 ppb. NO was injected into the chamber from a concentrated
NO cylinder source (NO = 133.16 ppm, balance $N_2$) to achieve an initial concentration of
~500 ppb. Seed aerosol was injected into the chamber by atomizing 0.06 M aqueous
ammonium sulfate solution to provide sufficient surface area for the partitioning of alkyl
nitrates (Nguyen et al., 2014a; Nguyen et al., 2014b; Zhang et al., 2014a; Zhang et al.,
2015b; McVay et al., 2016; Nah et al., 2016; Huang et al., 2018). The chamber contents
were allowed to mix for ~30 min before the onset of irradiation. After ~2 hr
photooxidation, NO was nearly depleted (>5 ppb) and the irradiation was ceased. SOA
produced was then collected on Teflon filters (47-mm diameter, 0.5-μm pore size,
MILLIPORE) through active sampling at a flow rate of 10 L/min for ~3 hr (Schilling
Fahnestock et al., 2014; Zhang et al., 2014b; Huang et al., 2016; Thomas et al., 2016).
Filters were stored in a -20 °C freezer prior to analysis (Riva et al., 2016). SOA samples
were extracted in 20 mL HPLC-grade methanol by 45 min of sonication at ~273 K and
then concentrated to ~5 mL with the assistance of a ~2 L/min $N_2$ stream.

## 2.3. Instrumental

The Electrospray Ionization Drift-Tube Ion Mobility Spectrometer (DT-IMS) interfaced to a Time-of-Flight Mass Spectrometer (TOFMS) was utilized in the characterization of ANs. The instrument was designed and manufactured by Tofwerk (AG, Switzerland), with detailed descriptions and schematics provided by previous studies (Kaplan et al., 2010; Groessl et al., 2015; Krechmer et al., 2016; Zhang et al., 2016b; Zhang et al., 2017). Here we will present the instrument operation protocols specific to the ANs measurement.

AN standards and SOA filter extracts were delivered to the ESI source via a 250 μL gas-tight syringe (Hamilton) held on a syringe pump (Harvard Apparatus) at a flow rate of 1 μL min$^{-1}$. The optimal ESI potential to readily generate stable ion adducts while minimizing the corona discharge was found to be $-1800$ V. The negatively charged mist generated at the emitter tip is introduced into the drift tube through a Bradbury-Nielson ion gate located at the entrance with the assistance of 1 L min$^{-1}$ nitrogen sheath gas. The BN ion gate was operated at the Hadamard Transform mode, with a closure voltage of 50 V and a gate pulse frequency of $1.2 \times 10^3$ Hz. The drift tube was held at a constant temperature (340±3 K) and atmospheric pressure (~766 *Torr*). A counter flow of N$_2$ drift gas was introduced at the end of the drift region at a flow rate of 1.2 mL min$^{-1}$. Ion mobility separation was carried out at the field strength ranging from 300 to 400 V cm$^{-1}$. After exiting from the drift tube, ions were focused into a pressure-vacuum interface that includes two segmented quadrupoles (Q$_1$ and Q$_2$) through an ion lens and a nozzle. Note that the potential gradient applied to the ion lens and nozzle should be limited to 500 V or less to prevent intensive fragmentation of the molecular ions. The frequency and amplitude were set as $1.5 \times 10^6$ Hz and 196 V for Q$_1$ and $1.5 \times 10^6$ Hz and 250 V for Q$_2$, respectively. Collision induced dissociation (CID) can be performed by adjusting the voltages on the ion optical elements between the two quadruple stages. Over the course of a CID program, the quadrupoles were set to $1.3 \times 10^6$ Hz and 120 V for Q$_1$ and $1.2 \times 10^6$ Hz and 150 V for Q$_2$, respectively, to ensure good transmission of low masses (*m/z* < 100).

The ESI-IMS-TOFMS instrument was operated in the *m/z* range of 20 to 1500 with a total recording time of 60 s for each dataset. The mass spectrometer was calibrated using sodium nitrate, ammonium phosphate, sodium dodecyl sulfate, sodium taurocholate hydrate, and ultramark 1621 in the negative mode. The ion mobility measurements were calibrated using tetrabutyl ammonium chloride as the instrument standard and 2,4-lutidine as the mobility standard (Zhang et al., 2016b). The average IMS ($t/dt_{50}$) and MS

($m/dm_{50}$) resolving powers are ~80 and ~4000, respectively. Mass spectra and ion
mobility spectra were collected by Aquility DAQ v2.1.0 and post processed by Tofware
v2.5.3.
**3. Results and Discussion**
**3.1. Ion adduct formation**
The strong electron affinity of the $-ONO_2$ functional group makes alkyl nitrate a
potential candidate for being analyzed in the negative electrospray ionization mode.
However, the ESI(−) mass spectra of the AN standards investigated here are typically
characterized by various fragments and clusters due principally to the thermally labile
$-ONO_2$ moiety. As shown in Figure 1, no molecular ion ($[M]^-$ or $[M\text{-}H]^-$) is observed
on the ESI(−) mass spectra of 1-mononitroglycerin (MNG), 1,3-dinitroglycerin (DNG),
and pentaerythritol tetranitrate (PETN). Instead, a small peak appears as a cluster ion of
the form $[M+NO_2\text{-}H]^-$. It is worth noting that addition of water to the mobile phase does
not promote the molecular ion formation, rather significant nitrate losses via hydrolysis
were observed. With the addition of trace amount of salts, i.e., ammonium chloride
($NH_4Cl$), sodium nitrate ($NaNO_3$), sodium iodide (NaI), and ammonium acetate ($NH_4Ac$),
the overall signal intensities were significantly enhanced through the production of a suite
of adduct ions of the form $[M+Cl]^-$, $[M+NO_3]^-$, $[M+I]^-$, and $[M+Ac]^-$, respectively.
The relative sensitivities of individual adduct ions increase by ultimately two orders of
magnitude, compared with the pure standard in methanol solution. Here the observed ion
adduct formation in ESI can be considered as a special case of chemical ionization
occurring in solution before the charge separation process takes place.
Table 1 lists the characteristic adduct ions formed from three AN standards (MNG,
DNG, and PETN) in methanol solution with selected additives ($NH_4Ac$, $NH_4Cl$, NaI, and
$NaNO_3$). Ion adducts are systematically observed from all of the ANs investigated,
regardless of the number of $-ONO_2$ functional groups attached on the molecule. Nitrate
($NO_3^-$) and chloride ($Cl^-$) anions were found to be the most effective additives to
promote ion adduct formation. Nitrate clusters exhibit the highest signal intensity and
lowest limit of detection, especially for the poly-nitrates and functionalized alkyl nitrates
investigated. Chloride clusters are characterized by two distinct ions with a mass
difference of 2 amu and abundance ratio of 3:1 due to the natural presence of isotopes
$^{35}$Cl and $^{37}$Cl. Also given in Table 1 are the detected negative ions from two organic nitro
compounds, i.e., hexahydro-1,3,5-trinitro-1,3,5-triazine (RDX) and 2,4-dinitrotoluene
(DNT). In contrast to RDX, which undergoes intensive clustering processes with $Cl^-$, $I^-$,
and $NO_3^-$ during negative ESI, one dominant molecular ion ($[M\text{-}H]^-$) was observed on
the ESI(−) mass spectra of DNT. The limits of detection (LOD) towards the nitrate
adducts are in the range of 0.1 to 4.3 µM (see Table 1), demonstrating an improved
performance of the IMS-MS technique employed here compared with literature data
obtained from sprayed solutions (Asbury et al., 2000; Hilton et al., 2010). For example,
the LODs for DNT and RDX are 26 µg/L and 40 µg/L, respectively, in Asbury et al.
(2000), and the LOD for urea nitrate is 2.5 mg/L in Hilton et al. (2010).

206       The effect of the additive concentrations ($NO_3^-$ and $Cl^-$) on the ion adduct formation

was investigated using an equimolar mixture (5 µM each) of PETN and RDX as
representative of nitrates and nitro compounds, respectively, in methanol solution (Figure
2). In the absence of any additives, the presence of background anions from either
impurities in the solvent or thermal decomposition of alkyl nitrates leads to a detectable
amount of ion adducts. With the anion levels on the order of micromolar, ion adducts
become dominant in the ESI(−) mass spectra. The optimal anion concentration was found
to be in the range of 0.01 mM to 0.1 mM. Progressively rising anion concentrations (>
1mM) essentially suppress adduct formation due to the competition for limited resources,
such as space and charge (Cech and Enke, 2001). Note that the measured drift time for
each ion adduct is constant at anion concentrations ranging from 1 µM to 1 mM,
indicative of the absence of ion-molecule clustering in the IMS drift tube.
**3.2. Collision cross section vs. mass to charge ratio trend line**

219       Collision cross section ($\Omega_{N_2}$) represents the effective area for interactions between a

charged molecule and the surrounding buffer gases (e.g., $N_2$ herein). It is derived from
the mobility measurement in the IMS drift tube, where ions with open conformation
undergo more collisions with buffer gas molecules and hence travel more slowly than the
compact ones (Shvartsburg et al., 2000). The measured $\Omega_{N_2}$ for organic nitrates and nitro
compounds given in Table 1 are in good agreement with previous reported values
obtained from experiments where the analytes were introduced into the IMS system from
the vapor phase (Kaur-Atwal et al., 2009; Kozole et al., 2015). Combination of collision
cross section with molecular mass (as denoted by mass to charge ratio, *m/z*) provides a
two-dimensional space for separation of species based on their size as well as geometry.
We have shown that species of the same chemical class (e.g., amines, alcohols, and
carboxylic acids) tend to situate as a narrow band and follow a unique trend line on the 2-
D space (Zhang et al., 2016b). Here we demonstrate the presence of a $\Omega_{N_2} - m/z$ trend
line for alkyl nitrates. Figure 3 shows that the measured $\Omega_{N_2}$ of the AN adducts,
regardless of the AN molecular structures and types of anions that promote the adduct
formation, appear along the $\Omega_{N_2} - m/z$ trend line predicted by the core model (deviations
less than 5.2%). Also shown here are the predicted $\Omega_{N_2} - m/z$ trend lines for *mono/multi*-
carboxylic acids and organic sulfates, which readily produce molecular ions via
deprotonation ($[M-H]^-$) during negative ESI. Alkyl nitrates can be distinguished from
carboxylic acids and sulfates based on their distinct collision cross sections vs. mass to
charge ratio relationship. Note that other important chemical classes of atmospheric
interest, such as amines, alcohols, aldehydes, and peroxides, are suitable for analysis in
the positive ESI and their trend lines are not given here.

### 3.3. Characteristic fragments upon collision-induced dissociation

Molecular structures of selected AN ion adducts were further probed with the
assistance of the collision-induced dissociation (CID) analysis, which was performed
after the drift tube but prior to the time-of-flight chamber. The resulting daughter ion
appears at the same drift time as the parent ion, allowing for a straightforward correlation
of any given ion with its fragments. As shown in Figure 4, the nitrate ion ($NO_3^-$) at *m/z* 62
is exclusively observed upon CID of the parent ion adducts formed from MNG, DNG,
and PETN by clustering with $Cl^-$, $NO_3^-$, and $Ac^-$. The $NO_3^-$ fragment resulting from
decomposition of the corresponding parent ion adduct can be well separated from that
originally added to the AN solution based on their entirely different ion mobilities (as
reflected by the measured drift time). Thus $NO_3^-$ is considered as a characteristic
fragment upon CID of the parent AN adduct ion and serves as a tracer to verify the
presence of the $-ONO_2$ functional group in unknown compounds.
The anions ($Cl^-$, $NO_3^-$, and $Ac^-$) that promote the clustering chemistry were not
observed upon CID of the parent AN adducts. Figure 5 shows the profiles of four ion
adducts, i.e., $[MNG+Cl]^-$, $[MNG+Ac]^-$, $[PETN+Cl]^-$, and $[PETN+I]^-$, as well as their
resulting fragments under a sequence of CID potential gradient. As expected, the
abundance of the transmitted parent ion adducts decreases as the CID voltage rises. $NO_3^-$
appears as the largest product ion, and its enhanced abundance with increasing CID
voltage is balanced by the decrease in signals of the corresponding parent ion adduct. $Cl^-$
and $Ac^-$ remain minor peaks over the entire range of displayed CID potential gradient.
Under low-energy collisions, the parent AN ion adduct principally follows two
fragmentation pathways, leading to either $Cl^-$/ $Ac^-$/$I^-$ with the neutral AN molecule or
the deprotonated AN molecular ion ($[M-H]^-$) via the neutral loss of HCl / HAc / HI. The
absence of $Cl^-$ and $Ac^-$ indicates higher gas-phase basicity of $Cl^- / Ac^-$ than $[M-H]^-$.
As a result, the mechanism yielding $[M-H]^-$ is the dominant fragmentation pathway of
AN ion adducts (with an exception for $[PETN+I]^-$). The resulting molecular ion $[M-H]^-$
decomposes promptly to $NO_3^-$ due to the presence of the fragile $R–ONO_2$ bond.

### 3.4. Application to isoprene SOA

The OH-initiated oxidation of isoprene produces a population of isoprene peroxy
radicals ($RO_2$), the fate of which depends on the level of nitric oxide. Under high-NO
conditions as performed in the chamber experiments here, $RO_2$ radicals preferentially
react with NO leading to major first-generation products including isoprene hydroxy
nitrates, among which the two β-hydroxy nitrates dominate the isomer distribution. Due
to the presence of a double bond, the hydroxy nitrate could undergo OH addition
followed again by reactions of $RO_2$ radicals with NO, leading to a spectrum of products,
of which some highly functionalized molecules such as the dihydroxy dinitrate are
potential SOA precursors (Wennberg et al., 2018).
A pair of ion adducts at $m/z$ 261 ($[M+^{35}Cl]^-$) and $m/z$ 263 ($[M+^{37}Cl]^-$) with the
abundance ratio of 3:1 is observed in the mass spectra of the isoprene SOA extracts in
methanol with 0.2 mM sodium chloride as the additive. These two adducts share an
identical mobility (DT = ~25.8 ms), which also appears as a small peak (DT = ~25.7 ms)
in the mobility spectra of the $NO_3^-$ ion (bottom panel of Figure 6). Further inspection of
the 'mobility-selected' mass spectra of the parent ion adduct at $m/z$ 261 reveals that $NO_3^-$
is the major fragment ion (top panel of Figure 6). With the application of a CID potential
sequence, the intensity of the precursor ion at $m/z$ 261 decreases and that of the fragment
ion at $m/z$ 62 increases (middle panel of Figure 6), a similar pattern observed for the AN
standards. We thereby tentatively assign the parent ion adduct at $m/z$ 261 to a second-
generation oxidation product, dihydroxy dinitrate ($C_5H_{10}O_8N_2$, see the chemical structure
given in Figure 6), which is produced from the addition of OH to the two double bonds of
isoprene followed by $RO_2+NO$ reactions. It is interesting to note that a small shoulder
peak appears at ~26.0 ms in the mobility spectra of the ion adduct at $m/z$ 261 (bottom
panel of Figure 6), likely representative of the $C_5H_{10}O_8N_2$ isomers generated from the
much less favored OH-addition channels that produce primary $RO_2$ radicals. Quantitative
analysis of the dihydroxy dinitrate is complicated by the matrix interference during the
ESI process and chromatographic separation prior to infusion to the ESI source is
required (Zhang et al., 2015a; Zhang et al., 2016a), which is beyond the capability of the
current instrument setup. Further note that first-generation hydroxy nitrates were not
detected, due to their relatively high volatility and thus quite limited partitioning onto the
particle phase. On the other hand, multiple peaks were observed in the mobility spectra of
the $NO_3^-$ ion (bottom panel of Figure 6), and their drift times are higher than that of the
ion assigned to the dihydroxy dinitrate, implying that some high-molecular-weight nitrate
products were likely fragmented in the quadrupole interface.

**4. Conclusions**

The anion attachment chemistry was previously used in the negative ESI operation to
effectively induce ion formation from neutral molecules that lack acidic sites (Zhu and
Cole, 2000). Here we build upon the use of anion attachment, a special chemical
ionization mechanism in solution, to characterize the condensed-phase alkyl nitrates at
molecular level.  The propensity of the $-ONO_2$ moiety to cluster with a diverse selection
of anions, including $Cl^-$, $I^-$, $NO_3^-$, and $Ac^-$, was observed during the negative
electrospray ionization process, and the measured total ion signals were enhanced by
ultimately two orders of magnitude. Compared with conventional mass spectrometric
techniques, the coupled ion mobility and mass-to-charge ratio measurements provide a
two-dimensional separation of alkyl nitrates from other chemical classes commonly
detected in negative ESI, such as organic sulfates and carboxylic acids. With the
assistance of the collision-induced dissociation analysis, upon which the resulting product
ions share the identical drift time as the precursor ion, molecular structures of ANs can be
further probed. Regardless of the types of anions attached to the AN molecules,
dissociation of the parent adduct ion yields a characteristic fragment, $NO_3^-$ at *m/z* 62,
which can be used to verify the presence of the $-ONO_2$ functional group in any given
molecule. These new features enable the unambiguous identification of alkyl nitrates in a
complex organic mixture, as exemplified by the detection of hydroxynitrates in isoprene
derived SOA. The IMS-MS technique for the measurement of condensed-phase ANs is in
its early stages of development. Accurate quantification of a given AN molecule by
minimizing the ion suppression and improving the long-term stability of ESI is needed
for future work.

**Acknowledgements**

The National Center for Atmospheric Research is operated by the University
Corporation for Atmospheric Research, under the sponsorship of the National Science
Foundation.

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

Table 1. Overview of compounds containing $-ONO_2$ and $-NO_2$ functional groups
investigated in this study.

| Compound | Molecular Formula | Ion | | LOD [a] ($\mu$M) | $\Omega_{N_2}$ [b] ($\text{Å}^2$) | Structure |
|---|---|---|---|---|---|---|
| | | Formula | *m/z* | | | |
| 1-Mononitroglycerin (MNG) | $C_3H_7NO_5$ | $[M+Cl]^-$ | 172.0 | 0.8 | 129.4 | |
| | | $[M+NO_2-H]^-$ | 182.0 | 0.7 | 132.7 | |
| | | $[M+Ac]^-$ | 196.0 | 0.3 | 139.2 | |
| 1,3-Dinitroglycerin (DNG) | $C_3H_6N_2O_7$ | $[M+Cl]^-$ | 217.0 | 1.1 | 151.1 | |
| | | $[M+NO_2-H]^-$ | 227.0 | 4.3 | 156.6 | |
| | | $[M+NO_3]^-$ | 244.0 | 0.6 | 151.7 | |
| | | $[M+I]^-$ | 308.9 | 0.8 | 177.0 | |
| Pentaerythritol tetranitrate (PETN) | $C_5H_8N_4O_{12}$ | $[M-H]^-$ | 315.0 | 1.1 | 181.7 | |
| | | $[M+Cl]^-$ | 351.0 | 0.5 | 183.7 | |
| | | $[M+NO_2-H]^-$ | 361.0 | 0.9 | 190.7 | |
| | | $[M+NO_3]^-$ | 378.0 | 0.2 | 190.9 | |
| | | $[M+I]^-$ | 442.9 | 0.1 | 216.2 | |
| | | $[2M+Cl]^-$ | 667.0 | 1.0 | 262.6 | |
| 2,4-Dinitrotoluene (DNT) | $C_7H_6N_2O_4$ | $[M-H]^-$ | 181.0 | 0.6 | 137.0 | |
| Hexahydro-1,3,5-trinitro-1,3,5-triazine (RDX) | $C_3H_6N_6O_6$ | $[M+Cl]^-$ | 257.0 | 0.3 | 149.8 | |
| | | $[M+NO_2-H]^-$ | 267.1 | 1.4 | 156.3 | |
| | | $[M+NO_3]^-$ | 284.0 | 0.2 | 160.8 | |
| | | $[M+I]^-$ | 348.9 | 0.1 | 181.9 | |
| | | $[2M+Cl]^-$ | 479.0 | 1.6 | 203.5 | |

[a] The limit of detection (LOD) is calculated as LOD = $\sigma \times$(S/N)/$k$, where S/N is the signal-to-noise ratio, which is taken as 3 here, $k$ is the response factor of IMS-MS towards individual ion adducts produced from 5 $\mu$M standard nitrate solution during negative ESI, and $\sigma$ is the standard deviation of the IMS-MS response over the course of 60 s measurements.

[b] The collision cross section ($\Omega_{N_2}$) is calculated through the modified zero field (so called Mason-Schamp) equation, see more details in Zhang et al. (2016).


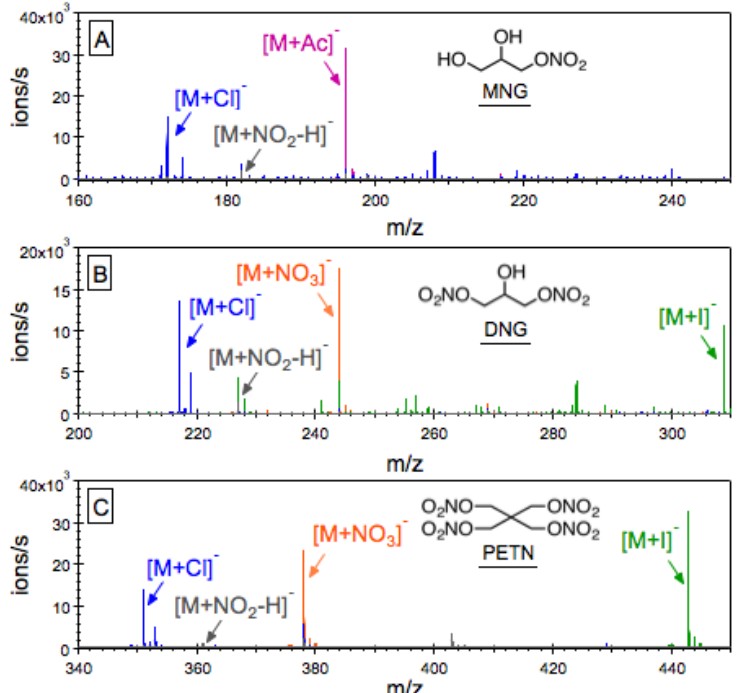



Figure 1. Negative ESI mass spectra of 5 µM 1-mononitroglycerin (MNG), 1,3-
dinitroglycerin (DNG), and pentaerythritol tetranitrate (PETN) dissolved in pure
methanol (gray), methanol with 0.1 mM ammonium acetate (NH$_4$Ac, purple), methanol
with 0.1 mM ammonium chloride (NH$_4$Cl, blue), methanol with 0.1 mM sodium nitrate
(NaNO$_3$, orange), and methanol with 0.1 mM sodium iodide (NaI, green). These three
alkyl nitrates, which do not readily produce significant amount of molecular ions on their
own during negative ESI, are observed as clusters with acetate (Ac$^-$), chloride (Cl$^-$),
nitrate (NO$_3^-$), and iodide anions (I$^-$) in the ESI($-$) spectra.





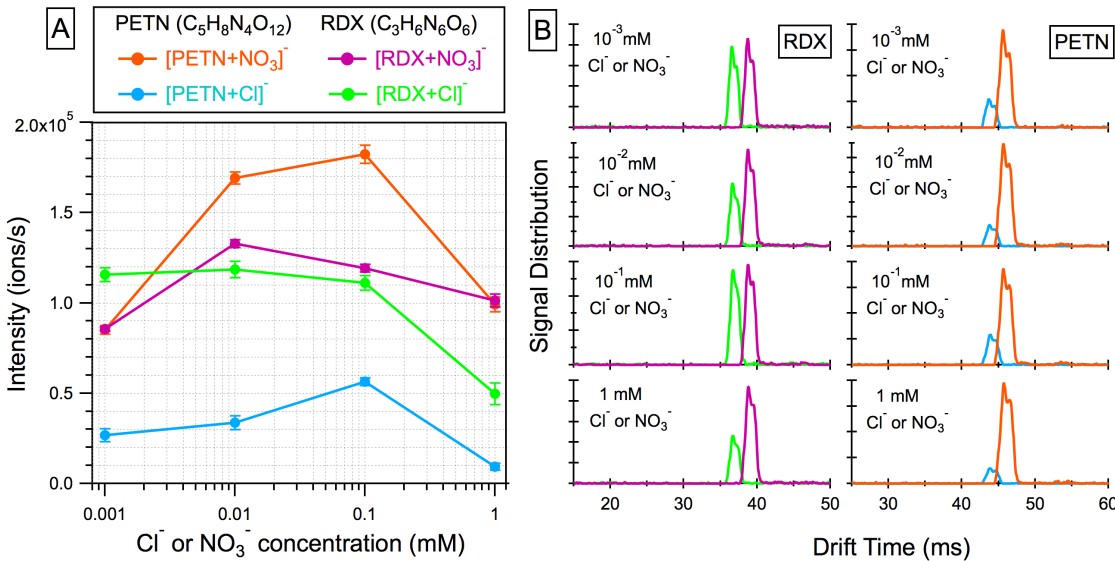



Figure 2. (A) Signals of the ion adducts produced from RDX and PETN by clustering
with chloride ($Cl^-$) and nitrate ($NO_3^-$) as a function of the corresponding anion
concentrations ranging from 1 μM to 1 mM. (B) Drift time distributions of the ion
adducts $[RDX+Cl]^-$, $[PETN+Cl]^-$, $[RDX+NO_3]^-$, and $[PETN+NO_3]^-$ are consistent at
different anion concentrations.







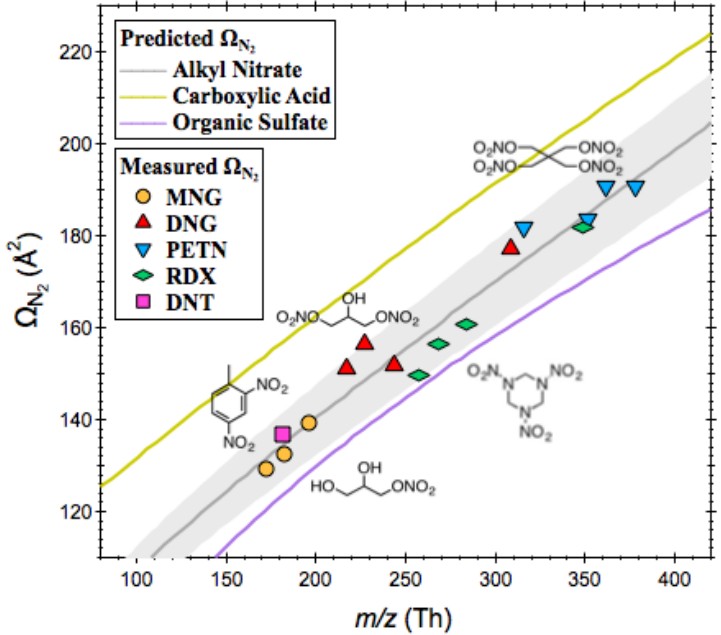



Figure 3. Measured collision cross sections ($\Omega_{N_2}$) of the AN ion adducts as a function of
their mass-to-charge ratios appear along the predicted $\Omega_{N_2} - m/z$ trend line. Also shown
here are the predicted $\Omega_{N_2} - m/z$ trend lines for carboxylic acids and organic sulfates,
which are major chemical classes of atmospheric interest detected in the negative ESI
mode.


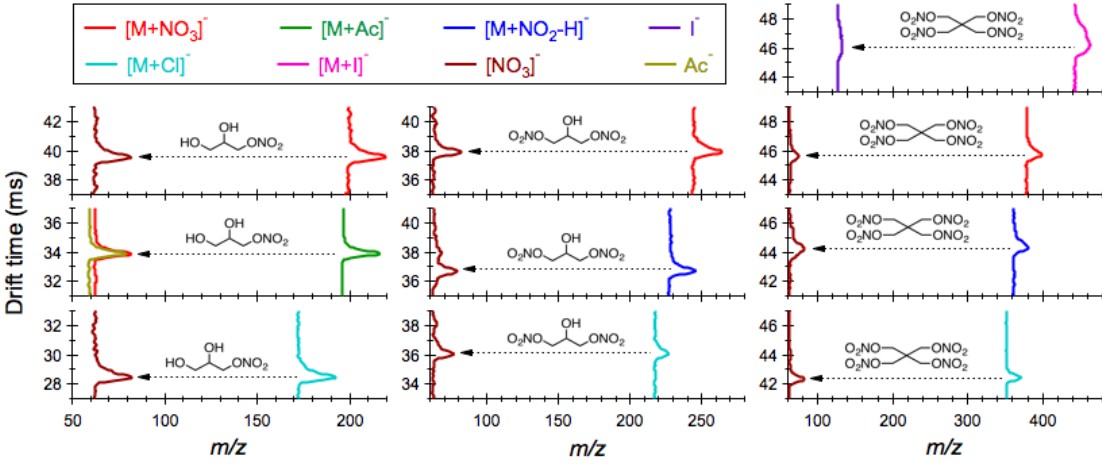

Figure 4. Characteristic fragment ions produced from MNG, DNG, and PETN by clustering with acetate (Ac⁻), chloride (Cl⁻), iodide (I⁻), nitrate ($NO_3^-$), and nitrite ($NO_2^-$) upon collision induced dissociation performed at a CID voltage of 20 V.


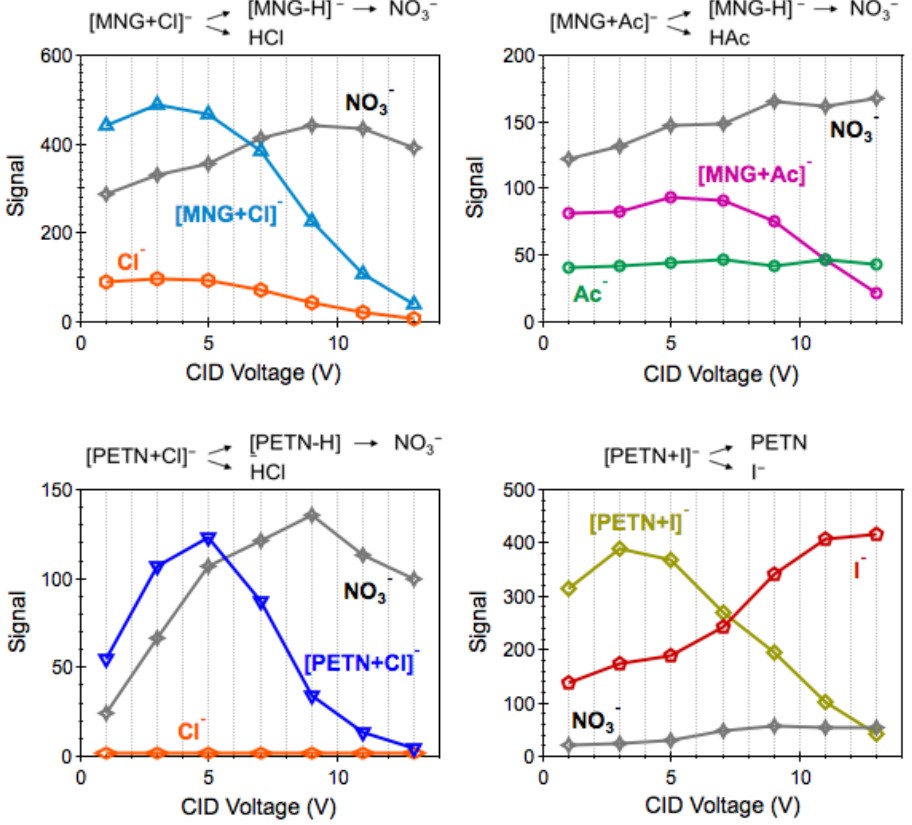



Figure 5. Peak intensities of the precursor ion adducts [MNG+Cl]⁻, [MNG+Ac]⁻,
[PETN+Cl]⁻, and [PETN+I]⁻ as well as their fragment ions as a function of the collision
energy as displayed by the CID voltage.








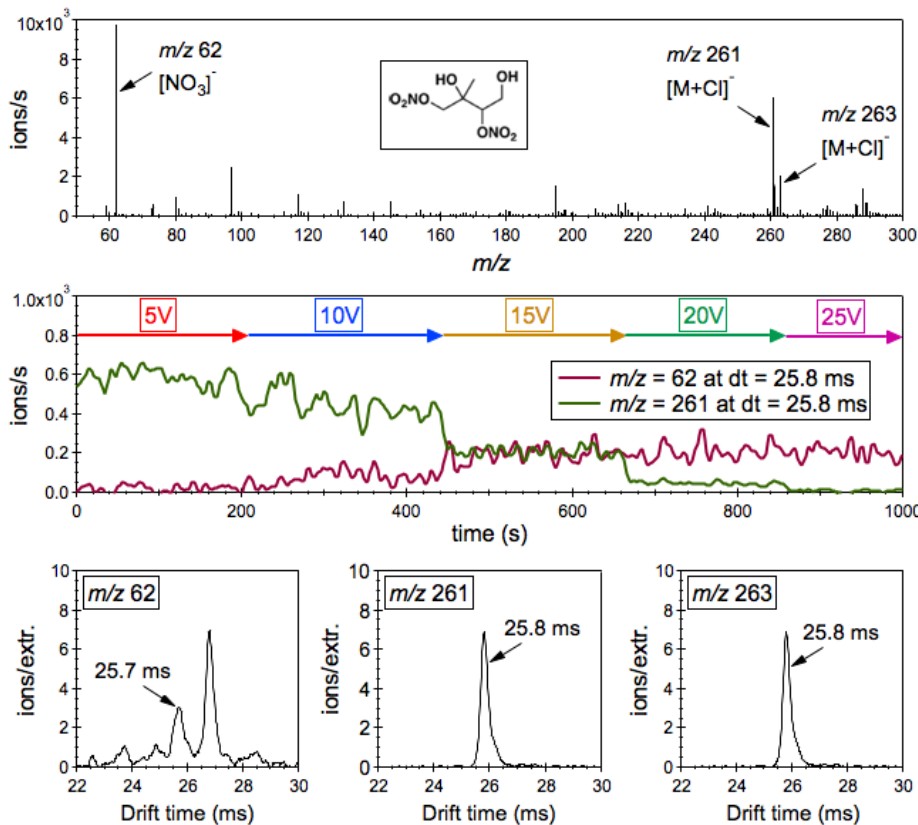



Figure 6. (Top panel) The 'mobility-selected' mass spectra of the parent ion adduct at *m/z*
261 and its major fragment at *m/z* 62 in isoprene SOA extracts with ~0.2 mM sodium
chloride as the additive. (Middle panel) Profiles of the precursor ion adduct at *m/z* 261
and its product ion at *m/z* 62 as a function of the CID voltage. (Bottom panel) Drift time
spectra of the ion adduct at *m/z* 261, its isotope ion adduct at *m/z* 263, and the fragment
ion at *m/z* 62.