# Peer review of "Molecular Characterization of Alkyl Nitrates in Atmospheric Aerosols by Ion Mobility Mass Spectrometry"

_Atmospheric Measurement Techniques, 2019_

## Referee Comment (RC1) · Anonymous Referee #1 · 8 Jul 2019

General Comment:

This is a short manuscript reporting the development of a new technique based on Ion Mobility Mass Spectrometry (IMS-MS) for the analysis of alkyl-nitrates in atmospheric and laboratory-generated aerosols. Although the analysis of water soluble organic compounds in atmospheric aerosols by coupling of IMS-MS with Electrospray Ionization (ESI) has already been demonstrated, this work presents the first development focusing on organic nitrates. This new technique addresses important challenges in atmospheric chemistry by aiming at separating the many alkylnitrate isomers present in atmospheric samples and identifying each of them by collision-induced dissociation.

[Figure]

Even if this analysis is only qualitative for now, it is very relevant for atmospheric chemistry as alkylnitrates are key tracers for atmospheric oxidation mechanisms but can be separated by only few techniques and are notoriously sensitive to decomposition.

One of the main findings in this work is that, while alkylnitrates alone do not ionize easily with ESI, their ionization is considerably enhanced by clusterization with anions such as chloride, nitrate, iodine, and acetate, thus providing a new way to detect them. The systematic work presented here led to an increase of the alkylnitrate ion signals by 2 orders of magnitude.

As underlined by the authors themselves, the technique is still in an early stage of development. For instance, its application to the analysis of secondary aerosols produced by the oxidation of isoprene in a reaction chamber did not allow to detect first-generation hydroxy-nitrates.

In spite of these limitations, this technique is very relevant for atmospheric chemistry, therefore I recommend the publication of this manuscript.

Minor comments:

1) Combining the analysis of volatile and non-volatile compounds

One of the main limitations underlined by the analysis of the SOA produced by the oxidation of isoprene is that first-generation hydroxyl-nitrates could not be detected due to their volatility. Previous applications of IMS-MS (for instance Krechmer et al. 2015, 2016) have used different ionizing sources to analyze both gas- and condensed-phase products. Although it might not be convenient to change the ionization source, could a similar strategy (or a different one) be used to detect all the alkylnitrates in a system ?

2) Quantification

The qualitative identification of different alkylnitrates with this technique is already very valuable, as alkylnitrates are tracers for specific reaction pathways. Thus this technique

could already lead to the identification of previously unknown pathways. But quantification would certainly be a plus. Would quantification be possible, for instance from the ion mobility spectra, in a similar way as in chromatographic techniques ?

3) There was a few minor mistakes in the text:

- p.5, Li. 126: shouldn't it be "with the assistance of..." instead of "with the assistant of..."

- p. 7, Li. 170, "ubiquitously" does not seem to be the right word here. "systematically" might be more appropriate.

- p. 7, Li. 192: it should probably be "constant" instead of "consistent". Or "the same at all anion concentration".

- p. 9, Li.252: the reference "Wennberg et al., 2018" is not in the reference list, please check.

---

## Referee Comment (RC2) · Anonymous Referee #2 · 15 Aug 2019

The manuscript presents an application of Ion mobility mass spectrometry to alkyl nitrate characterization. The application is new for atmospheric chemistry experiments. The experiments are sound and well carried out. The results are well presented and discussed and I recommend publication of the manuscript after the following issues have been addressed.

Major issues:

The main issue is novelty and a very biased representation of the literature. In fact, the alkyl nitrates the authors target are explosives and their analysis by Ion mobility spectrometry has been studied by the security community for over a decade! So ig-

noring this work is not appropriate! It i critical to actually include this and discuss what is novel here? What insights that are not already published have been gained and mostly how does this work relate in terms of selectivity and sensitivity to the existing literature. Even high resolution ESI-HRIMS work has been published nearly 10 years ago on RDX and similar compounds. It is amazing that this work (Hilton et al., 2010) is not referenced and discussed here. Also many other open literature papers are out there on "nitro explosive" detection which is the same as the alkylnitrates mentioned here.

A second, partially related issue that is critical to address is a lack of figures of merit in the abstract and very little to no discussions of figures of merit in the text. For an analytical paper it would be customary to have quantitative information in the abstract (e.g. LOD) and discussed in the text.

Relevant papers

Hilton C.K., Krueger C.A., Midey A.J., Osgood M., Wu J., Wu C. Improved analysis of explosives samples with electrospray ionization-high resolution ion mobility spectrometry (ESI-HRIMS) Int. J. Mass. Spectrom. 298, 64–71, 2010.

Kozole, J. et al., Gas phase ion chemistry of an ion mobility spectrometry based explosive trace detector elucidated by tandem mass spectrometry, Talanta, 140, 10-19, 2015.

But also:

Sivakumar N. et al., Development of an ion mobility spectrometer for detection of explosives. Instrum. Sci. Technol. 41:96–108, 2013.

And quite a few others. . . just replace alkyl nitrate with nitro explosives (same compounds!). . .

Details:

Please provide source information consistently for all your chemicals (target compounds).

Please provide quantitative information in the abstract (LODs) and also provide a comparison to the existing explosive literature.

Figure 2: Collision cross section typically has a unit (Aˆ2?).

---

## Author Comment (AC1) · 7 Sep 2019

**Responses to Reviewer #2**

The manuscript presents an application of Ion mobility mass spectrometry to alkyl nitrate characterization. The application is new for atmospheric chemistry experiments. The experiments are sound and well carried out. The results are well presented and discussed and I recommend publication of the manuscript after the following issues have been addressed.

*We thank reviewer #2 for the constructive comments. Our point-by-point responses can be found below, with reviewer comments in **black**, our responses in **blue**, alongside the relevant revisions to the manuscript in **red**.*

Major issues:

The main issue is novelty and a very biased representation of the literature. In fact, the alkyl nitrates the authors target are explosives and their analysis by Ion mobility spectrometry has been studied by the security community for over a decade! So ignoring this work is not appropriate! It is critical to actually include this and discuss what is novel here? What insights that are not already published have been gained and mostly how does this work relate in terms of selectivity and sensitivity to the existing literature. Even high resolution ESI-HRIMS work has been published nearly 10 years ago on RDX and similar compounds. It is amazing that this work (Hilton et al., 2010) is not referenced and discussed here. Also many other open literature papers are out there on "nitro explosive" detection which is the same as the alkylnitrates mentioned here.

[Response]

The reviewer has raised a very good point that the two terms 'explosives' and 'alkyl nitrates' are representative of the same class of organic compound that contains the -$ONO_2$ functionality. Here we would like to emphasize that the IMS-based method used in the present study is unique and inherently different from the classic IMS technique for explosive detection that reached its maturity between the late XX century and early XXI century.

We first clarify that the vast majority of IMS trace explosives detectors used for security screening are gas sampling and analysis instruments (e.g., Ewing et al., 2001; Eiceman and Stone, 2004; Cotte-Rodríguez et al., 2005; Fernandez-Maestre et al., 2010). It requires the ionization of sample explosives which is accomplished by a radioactive source such as Ni or Am. As a matter of fact, measurements of gas-phase alkyl nitrates in the atmosphere have also been

routinely performed by a suite of techniques such as chemical ionization mass spectrometer and laser-induced-fluorescence spectroscopy (see the introduction section in more detail).

The second clarification is the application of the electrospray ionization (ESI) technique to the explosive detection. In fact, the majority of works published actually employed the ESI derivatives, that is, the Desorption Electrospray Ionization (DESI) to detect trace amounts of explosives on ambient surfaces (e.g., Takats et al., 2004 and 2005; Cotte-Rodriguez et al., 2005 and 2006; Popov et al., 2005; Justes et al., 2007; Forbes et al., 2013; Morelato et al., 2013) or the Secondary Electrospray Ionization (SESI) for gas-phase explosive analysis (e.g., Tam et al., 2004; Martínez-Lozano et al., 2009; Bean et al., 2011; Vidal-de-Miguel et al., 2012; Crawford et al., 2013; Aernecke et al., 2015). The present study is focused on the direct analysis of liquid samples using ESI. The ionization process in sprayed liquids is mechanistically different from that in DESI or SESI.

The characterization of alkyl nitrates at molecular level in the condensed/liquid phase remains one of the analytical challenges in atmospheric chemistry due to the thermally labile properties of alkyl nitrates. The IMS-method we proposed in this study is capable of detecting alkyl nitrates as intact molecules in the condensed phase. The qualitative identification of different alkyl nitrates with this technique is already very valuable, as alkyl nitrates are tracers for specific reaction pathways. It is worth noting that existing literature for direct explosive detection in the liquid phase is rather limited. We appreciate the reviewer's suggestion on the Hilton et al. (2010) reference. But we need clarify that this study relies on the detection the $NO_3^-$ fragments of the explosives. For example, Figure 2 in Hilton et al. (2010) shows that the intensities of molecular ions are much lower than the resulting $NO_3^-$ fragments in the ion mobility spectra of several types of nitrate explosives measured in the negative ESI. This is consistent with our observations that the nitrate explosives produce mostly the $NO_3^-$ fragments at *m/z* 62 but very little molecular ions on their own, see the figure given below. Using the $NO_3^-$ fragment as an indicator of large alkyl nitrate molecules significantly limits the capability of identifying unknowns in atmospheric aerosols.

The novelty of the present study is that we have enabled the characterization of intact molecules of alkyl nitrates with minimal fragmentation in the liquid phase by systematically promoting the formation of ion-adducts (e.g., $[M+Cl]^-$, $[M+I]^-$, $[M+Ac]^-$, and $[M+NO_3]^-$) in negative ESI, as opposed to simply relying on the detection the $NO_3^-$ fragments of the explosives conducted in the majority of previous studies (including both gas and condensed phases). By adding anions in the sprayed solution to promote the formation of nitrate adducts, we could achieve increased sensitivity and improved specificity for atmospheric aerosol analysis.

[Figure]

The figure above shows that the nitrate explosives produce mostly $NO_3^-$ fragments but very little molecular ions on their own if directly infused into negative electrospray.

Another advantage of our method is that the detection limits are greatly improved by producing ion adducts of alkyl nitrates in negative ESI. For example, the detection limit of urea nitrate, the only compound that was quantitatively analyzed in Hilton et al. (2010), was 0.0025 $\mu g \; \mu L^{-1}$, which equals to 20 $\mu M$. As given in Table 1 in the present study, the detection limits of 1-mononitroglycerin (MNG), 1,3-dinitroglycerin (DNG), pentaerythritol tetranitrate (PETN), hexahydro-1,3,5-trinitro-1,3,5-triazine (RDX), and 2,4-dinitrotoluene (DNT) are in the range of 0.1 to 4.3 $\mu M$, an order of magnitude lower at least than the data obtained by Hilton et al. (2010).

Indeed, the alkyl nitrate standards used in the present study for characterizing the performance of the IMS-MS instrument are essentially explosives, and using IMS for the detection of explosive vapors has been a very classic and commercialized technique for many decades. However, the focus of the present study is to measure the nitrates as intact molecules in the liquid phase by systematically promoting the formation of ion adducts in negative electrospray, as opposed to the majority of existing studies that rely on detecting the fragments of alkyl nitrates, which significantly limits the sensitivity and specificity. While we use explosive standards to demonstrate the IMS-ESI-MS capability of detecting the $-ONO_2$ group, we did also apply this technique for the first time to characterize the alkyl nitrates in secondary organic

aerosols produced from the photooxidation of isoprene, the largest non-methane hydrocarbon emissions globally.

[Revisions]

"In this study, we present the first demonstration of the Ion Mobility Mass Spectrometry (IMS-MS) interfaced with an Electrospray Ionization (ESI) source that enables the molecular characterization of alkyl nitrates in atmospheric aerosols. The IMS technique has been widely employed in the fields of biochemistry and homeland security. The majority of previous studies that adapted ESI for IMS analysis employed either the Desorption Electrospray Ionization (DESI) to detect trace amounts of ANs on ambient surfaces (Cotte-Rodríguez et al., 2005; Popov et al., 2005; Takáts et al., 2005; Justes et al., 2007) or the Secondary Electrospray Ionization (SESI) for gas-phase ANs measurements (Tam and Hill, 2004; Martínez-Lozano et al., 2009; Crawford and Hill, 2013). The analysis of ANs directly from liquid solutions, on the other hand, has not yet been widely explored. Hilton et al. (2010) found that the $NO_3^-$ fragment dominates the IMS spectra of several types of ANs measured in the negative ESI, suggesting these nitrate molecules readily fragment due to the thermally labile nature of the $-ONO_2$ functionality, thereby resulting in the loss of molecular information of the targeted compounds. Here we show that with the addition of selected anions including chloride, nitrate, iodide, and acetate into the sprayed solution, molecular structures of ANs are largely maintained by producing ion clusters of the form $[M+Cl]^-$, $[M+NO_3]^-$, $[M+I]^-$, and $[M+Ac]^-$, respectively."

A second, partially related issue that is critical to address is a lack of figures of merit in the abstract and very little to no discussions of figures of merit in the text. For an analytical paper it would be customary to have quantitative information in the abstract (e.g. LOD) and discussed in the text.

[Response]

We reported the detection limits in Table 1 for all the compounds studied. As the reviewer suggested, we have added discussions of figures of merit and comparisons with existing literature in the abstract and the main text, also see the revisions given below.

[Revisions]

"We show significantly enhanced sensitivity towards the intact molecules of ANs by ultimately two orders of magnitude with the addition of inorganic anions such as chloride and nitrate to the negative sprayed solution."

"This approach enables the measurement of ANs that have low tendency to form molecular ions on their own with improved limit of detection in the range of 0.1 to 4.3 µM."

"Application of the IMS-MS technique is exemplified by the identification of hydroxy nitrates (with ~80 IMS resolving power and ~4000 MS resolving power) in secondary organic aerosols produced from isoprene photochemistry."

"The limits of detection (LOD) towards the nitrate adducts are in the range of 0.1 to 4.3 µM (see Table 1), demonstrating an improved performance of the IMS-MS technique employed here compared with literature data obtained from sprayed solutions (Asbury et al., 2000; Hilton et al., 2010). For example, the LODs for DNT and RDX are 26 µg/L and 40 µg/L, respectively, in Asbury et al. (2000), and the LOD for urea nitrate is 2.5 mg/L in Hilton et al. (2010)."

"The measured $\Omega_{N_2}$ for organic nitrates and nitro compounds given in Table 1 are in good agreement with previous reported values obtained from experiments where the analytes were introduced into the IMS system from the vapor phase (Kaur-Atwal et al., 2009; Kozole et al., 2015)"

Relevant papers

Hilton C.K., Krueger C.A., Midey A.J., Osgood M., Wu J., Wu C. Improved analysis of explosives samples with electrospray ionization-high resolution ion mobility spectrometry (ESI-HRIMS) Int. J. Mass. Spectrom. 298, 64–71, 2010.

[Response] Have cited this paper as suggested.

Kozole, J. et al., Gas phase ion chemistry of an ion mobility spectrometry based explosive trace detector elucidated by tandem mass spectrometry, Talanta, 140, 10-19, 2015.

[Response] This is for gas phase explosive analysis and irrelevant to this study.

But also:

Sivakumar N. et al., Development of an ion mobility spectrometer for detection of explosives. Instrum. Sci. Technol. 41:96–108, 2013.

[Response] This is for gas phase explosive analysis and irrelevant to this study.

And quite a few others... just replace alkyl nitrate with nitro explosives (same compounds!)…

[Response] There have been numerous studies on the gas-phase characterization of explosives by IMS that utilized thermal desorption of a sample in conjunction with a radioactive ionization source. These studies are irrelevant to the focus of the present study that characterizes explosives/nitrates in the liquid phase (at least the ionization methods are entirely different).

Details

Please provide source information consistently for all your chemicals (target compounds).

[Response] Revised as suggested.

[Revisions]

"Organic nitrate and nitro standards stored in acetonitrile ampules, including 1-mononitroglycerin (100 µg/mL, SigmaAldrich), 1,3-dinitroglycerin (100 µg/mL, SigmaAldrich), pentaerythritol tetranitrate (1000 µg/mL, SigmaAldrich), hexahydro-1,3,5-trinitro-1,3,5-triazine (1000 µg/mL, SigmaAldrich), and 2,4-dinitrotoluene (1000 µg/mL, SigmaAldrich), were further diluted with methanol (HPLC grade, J. T. Baker) to 5 µM or less for characterizing the performance of the Ion Mobility Mass Spectrometer. Stock solutions of ammonium acetate (>99%, SigmaAldrich), ammonium chloride (>99%, SigmaAldrich), sodium nitrate (>99%, SigmaAldrich), and sodium iodide (>99%, SigmaAldrich) were prepared at a concentration of 10 mM in methanol. They were used as additives at typical concentrations of 0.01 – 0.1 mM in the ANs methanol solutions to promote the ion adducts formation."

Please provide quantitative information in the abstract (LODs) and also provide a comparison to the existing explosive literature.

[Response] Revised as suggested, also see revisions given in our earlier responses.

Figure 2: Collision cross section typically has a unit (A^2?).

[Response] That is correct.

---

## Author Comment (AC2) · 7 Sep 2019

**Responses to Reviewer #1**

General Comment:

This is a short manuscript reporting the development of a new technique based on Ion Mobility Mass Spectrometry (IMS-MS) for the analysis of alkyl-nitrates in atmospheric and laboratory-generated aerosols. Although the analysis of water soluble organic compounds in atmospheric aerosols by coupling of IMS-MS with Electrospray Ionization (ESI) has already been demonstrated, this work presents the first development focusing on organic nitrates. This new technique addresses important challenges in atmospheric chemistry by aiming at separating the many alkylnitrate isomers present in atmospheric samples and identifying each of them by collision-induced dissociation.

Even if this analysis is only qualitative for now, it is very relevant for atmospheric chemistry as alkylnitrates are key tracers for atmospheric oxidation mechanisms but can be separated by only few techniques and are notoriously sensitive to decomposition.

One of the main findings in this work is that, while alkylnitrates alone do not ionize easily with ESI, their ionization is considerably enhanced by clusterization with anions such as chloride, nitrate, iodine, and acetate, thus providing a new way to detect them. The systematic work presented here led to an increase of the alkylnitrate ion signals by 2 orders of magnitude.

As underlined by the authors themselves, the technique is still in an early stage of development. For instance, its application to the analysis of secondary aerosols produced by the oxidation of isoprene in a reaction chamber did not allow to detect first- generation hydroxy-nitrates.

In spite of these limitations, this technique is very relevant for atmospheric chemistry, therefore I recommend the publication of this manuscript.

*We thank reviewer #1 for the constructive comments. Our point-by-point responses can be found below, with reviewer comments in **black**, our responses in **blue**, alongside the relevant revisions to the manuscript in **red**.*

1) Combining the analysis of volatile and non-volatile compounds

One of the main limitations underlined by the analysis of the SOA produced by the oxidation of isoprene is that first-generation hydroxyl-nitrates could not be detected due to their volatility. Previous applications of IMS-MS (for instance Krechmer et al. 2015, 2016) have used different

ionizing sources to analyze both gas- and condensed- phase products. Although it might not be convenient to change the ionization source, could a similar strategy (or a different one) be used to detect all the alkylnitrates in a system?

[Respons] Krechmer et al. (2016) employed the same IMS instrument for the SOAS field campaign as the one used in the current study, but with different ionization source, i.e. the $NO_3$-ionization source for the gas-phase measurement. However, there is a major challenge in coupling the $NO_3$-ionization inlet to the IMS chamber, that is, the voltage applied to the $NO_3$-ionization inlet needs to be over 10K volts in order to push the ions through the entrance of the IMS, thereby significantly limiting the ionization efficiency of analytes of interest. To our knowledge, this technique is no longer pursued by the manufacturer. The electrospray ionization method, on the other hand, has no such limitation as it inherently requires high voltage to produce ions and/or ion clusters. We think one potential strategy to detect all the alkylnitrates in a system is to combine the secondary electrospray ionization (SESI) for the analysis of trace concentrations of vapors with the extracted electrospray ionization (EESI) for the analysis of particle-phase constituents.

2) Quantification

The qualitative identification of different alkylnitrates with this technique is already very valuable, as alkylnitrates are tracers for specific reaction pathways. Thus this technique could already lead to the identification of previously unknown pathways. But quantification would certainly be a plus. Would quantification be possible, for instance from the ion mobility spectra, in a similar way as in chromatographic techniques?

[Response] The quantitative characterization of alkylnitrates by IMS-MS is limited by the ESI process because one critical issue in quantitative ESI is the suppression of ionization due to matrix interference. For example, a particle filter sample would give significantly lower ionization signals compared to pure standard solutions with similar analyte concentrations. To overcome the matrix interference, separation such as adding a liquid chromatography column prior to the entrance of the ESI unit is desired. The resulting spectra should be something similar to the sketch given below: for any given peak on the chromatograph, there are correspondingly one plot of ion mobility spectra and one plot of mass spectra showing the major ion (analyte) constituting this chromatographic peak.

[Figure]

3) There were a few minor mistakes in the text:

    - p.5, Li. 126: shouldn't it be "with the assistance of. . ." instead of "with the assistant of..."

[Response] Revised as suggested.

    - p. 7, Li. 170, "ubiquitously" does not seem to be the right word here. "systematically" might be more appropriate.

[Response] Revised as suggested.

    - p. 7, Li. 192: it should probably be "constant" instead of "consistent". Or "the same at all anion concentration".

[Response] Revised as suggested.

    - p. 9, Li.252: the reference "Wennberg et al., 2018" is not in the reference list, please check.

[Response] Revised as suggested.